# Evaluation of DNA-Damaging Effects Induced by Different Tanning Agents Used in the Processing of Natural Leather—Pilot Study on HepG2 Cell Line

**DOI:** 10.3390/molecules27207030

**Published:** 2022-10-18

**Authors:** Sanja Ercegović Ražić, Nevenka Kopjar, Vilena Kašuba, Zenun Skenderi, Jadranka Akalović, Jasna Hrenović

**Affiliations:** 1Department of Materials, Fibres and Textile Testing, University of Zagreb Faculty of Textile Technology, Prilaz Baruna Filipovića 28a, 10000 Zagreb, Croatia; 2Mutagenesis Unit, Institute for Medical Research and Occupational Health, Ksaverska Cesta 2, 10000 Zagreb, Croatia; 3Division of Biology, Faculty of Science, University of Zagreb, Rooseveltov Trg 6, 10000 Zagreb, Croatia

**Keywords:** comet assay, %DNA in tail, genotoxicity, in vitro, tanning, natural leather

## Abstract

For a long time, the production and processing of cowhide was based on the use of chrome tanning. However, the growing problem with chromium waste and its negative impact on human health and the environment prompted the search for more environmentally friendly processes such as vegetable tanning or aldehyde tanning. In the present study, we investigated the DNA-damaging effects induced in HepG2 cells after 24 h exposure to leather samples (cut into 1 × 1 cm^2^ rectangles) processed with different tanning agents. Our main objective was to determine which tanning procedure resulted in the highest DNA instability. The extent of treatment-induced DNA damage was determined using the alkaline comet assay. All tanning processes used in leather processing caused primary DNA damage in HepG2 cells compared to untreated cells. The effects measured in the exposed cells indicate that the leaching of potentially genotoxic chemicals from the same surface is variable and was highest after vegetable tanning, followed by synthetic tanning and chrome tanning. These results could be due to the complex composition of the vegetable and synthetic tanning agents. Despite all limitations, these preliminary results could be useful to gain a general insight into the genotoxic potential of the processes used in the processing of natural leather and to plan future experiments with more specific cell or tissue models.

## 1. Introduction

Natural leather is a unique biological material with a complex morphological structure, the processing of which is extremely complex and requires many process steps, the most important of which is tanning, through which its characteristic and unique physical, chemical and mechanical properties are obtained. The main components of natural leather are proteins—collagen, keratin, elastin and reticulin—which make up about 33% of leather. Collagen, as the main component of leather, is responsible for the strength and toughness of raw and finished leather. Keratin is the only protein found in hair and the second most abundant protein in leather. Elastin is found in the papillary layer and is removed when the leather is processed. Besides collagen, proteoglycans, hyaluronic acid and smaller amounts of carbohydrates are also present in the reticular layer. Lipids are present in both the papillary and the reticular layers, the concentration depending on the origin of the leather (e.g., bovine leather contains 2–6% lipids) [1,2]. Collagen fibrils form a complex network and their arrangement contributes significantly to the strength of collagen structures [3]. The fibrils are grouped into fibril bundles that form collagen fibres, and the fibres fuse together to form bundles of collagen fibres [4]. This hierarchical part of the collagen fibre is important because it depends on the opening of the fibre structure during preparation for the tanning process. The splitting of the fibre structure at this stage of processing is important to achieve the final softness and strength of the finished hide [5].

The finished hide is a product made by processing raw animal hide in the tanning process by various means, the most common of which is the chrome tanning process, which makes the hide more durable and supple, and prevents rotting. The tanning agents used in the tanning process react chemically with the collagen molecule that makes up the natural skin and stabilise the triple helix structure of the collagen nucleus, making the leather resistant to chemical, thermal and microbiological degradation. The tanning agents used are usually inorganic compounds such as chromium, aluminium, iron, titanium and organic compounds such as aldehydes and vegetable tannins, synthanes and combinations of these compounds [6]. It is very important that the tannins used have the affinity and ability to react with collagen and have the appropriate size to penetrate the collagen fibres and achieve the desired cross-linking of the structure [7].

The manufacture and processing of leather has been based mainly on the use of chrome tanning salts for the last two centuries. Chromium salts are involved in the processing of more than 90% of the leather produced worldwide [8]. However, the growing problem of chromium waste and its negative impact on human health and the environment is prompting a search for more environmentally friendly methods. Several alternative processes to chrome tanning, such as vegetable tanning or aldehyde tanning [9,10,11], have been developed to produce chrome-free leather. Although these processes are gradually gaining commercial importance, these materials often cannot match the properties of chrome-tanned leather.

Considering the above, in this study we wanted to assess how the tanning procedures used differ according to their toxic potential. As known, for each new material or newly developed technological process various safety issues must be known before it is put into widespread use. Otherwise, it is possible that the material could cause unwanted health effects in the end users. 

Generally, the first step in the evaluation of potential toxic effects of unknown or poorly known mixtures of chemicals is selection of a suitable test model. Testing strategies often are complex; they should be performed in some logical sequence, and usually start with the in vitro assays. In further steps, depending on the preliminary results, investigations extend to the higher levels of biological organisation, where various in vivo models are used.

Nowadays, in vitro testing relies on the use of different cell lines. Each of them has its own specific characteristics, according to which the best model for testing and evaluating the effects could be selected. For the purpose of the present study, we selected the HepG2 cell model, which has already proven useful for studying the genotoxicity of many direct and indirect mutagens and compounds with unknown or poorly known mechanisms of action [12,13]. As HepG2 cells were successfully used in several previous studies which investigated toxic potential of various dyes and agents common in the textile and leather industry [14,15,16,17,18,19], it seems reasonable to select them for this experiment. We considered HepG2 cells a suitable test system because they retain—to a certain extent—the activity of metabolic enzymes that are important for the biotransformation of chemicals in the liver, and exhibit many of the genotypic and phenotypic characteristics of liver cells [20,21,22]. Keeping these facts in mind, for the purposes of the pilot experiment we first selected that specific cell model. The logic of such a choice was to assess whether there will be measurable DNA damaging effects on the HepG2 cell model, and in accordance with the obtained results, to propose directions for further research. This research was intended as a preliminary assessment of the DNA-damaging potency of different procedures in leather processing and their possible harmful effects on DNA integrity. We investigated the DNA-damaging effects induced in HepG2 cells after 24 h of exposure to leather samples processed with different tanning procedures. This research starts from a hypothesis that different tanning procedures could result in varying degrees of DNA damage. In line with this, our main objective was to determine which tanning process resulted in the highest DNA instability in the model cells. For this reason, the alkaline comet assay (single cell gel electrophoresis) was applied, which is considered one of the elementary tests for screening and early assessment of DNA-damaging effects. It enables detection of primary DNA damage at the level of an individual cell, inflicted by direct action of various chemical (or physical) agents or by indirect action of free radicals (for example, reactive oxygen species). The method was named after the appearance of the pattern of damaged DNA, which, stained with specific dyes and observed under a fluorescence microscope, resembles a celestial body—“comet”. The method constists of several steps, starting with embedding of cells in agarose microgel, lysis of their cytoplasm and membrane structures, denaturation with alkaline buffer—which potentiates removal of histones—strand separation and release of relaxed DNA loops. During electrophoresis, DNA loops migrated from the comet head towards the anode. Their migration pattern can be visualised after staining with fluorescent dyes that bind to the DNA. The measurement of comets can be performed under a fluorescence microscope using image analysis software or by manual scoring. The most important descriptors of DNA damage are “tail intensity” (the percentage of DNA that has migrated into the comet tail) and “tail length” (the length of DNA migration expressed in μm). The measured data are automatically stored in the form Microsoft Excel sheets, and what follows is further mathematical and statistical processing. The alkaline comet assay identifies a broad spectrum of lesions: Single and double-strand breaks in DNA, alkali-labile sites, single-strand breaks associated with incomplete excision repair, DNA–DNA or DNA–protein cross-links [23,24,25,26].

Based on all previously mentioned facts, we anticipate that the results obtained using the proposed in vitro experimental design could contribute new basic knowledge useful for the safety assessment of the studied tanning agents, which is important to identify potentially harmful tanning processes and consequently ensure the safety of leather products for consumers.

## 2. Results

After 24 h of incubation of HepG2 cells with leather samples, no changes in pH values were detected in the RPMI 1640 medium compared with the negative control sample.

The results of the alkaline comet assay (Figure 1) show statistically significant increases in the levels of %DNA in the tail (A) and tail length (B) in the treated HepG2 cells after 24 h of exposure to the tested leather samples compared to the negative control (*p* < 0.05; Mann–Whitney U test). The highest level of DNA damage was observed after exposure to the leather sample processed with vegetable tanning (label VEG-T), followed by the synthetic tanning (label SYN-T), and two chrome tanning procedures (labels CHR-T1 and CHR-T2). The levels of both comet descriptors determined in the positive control were significantly increased compared to all other experimental groups.

Typical appearances of the comets corresponding to different degrees of DNA damage are shown in Figure 2.

Further evaluations of DNA damage included analysis of the frequency distribution of comets measured in treated and control cells (Figure 3). The Y-axis represents the number of comets belonging to a particular category in relation to four quartiles: <25th percentile, 25th–50th percentile, 50th–75th percentile and >75th percentile. The differences between the treated cells and the control were tested using Pearson’s χ^2^ test, and the statistically significant differences compared with the negative control are shown in the above figure. The pattern of DNA damage measured in HepG2 cells after exposure to the tested samples of leather shows that for both comet descriptors (%DNA in tail and tail length), the proportion of comets belonging to the <25th percentile decreased significantly compared to the untreated cells, while the proportion of those belonging to the 75th percentile decreased significantly. Although there were some minor variations in the percentages of comets belonging to a particular category with respect to the four quartiles, none of the differences observed after exposure to the leather sample obtained by chrome tanning (label CHR-T2) was statistically significant compared to the negative control.

## 3. Discussion

The results of this pilot study confirmed that all tanning methods used in leather processing lead to primary DNA damage in HepG2 cells measurable by the alkaline comet assay. We also documented that “standard” chrome tanning, synthetic tanning and vegetable tanning differed in their ability to produce DNA lesions in HepG2 cells. Before going into detail about the significance of our results, we must point out that the available literature does not contain information about potentially DNA-damaging effects of tanning processes, nor does it provide recommendations on how to test them for their potentially harmful effects. Therefore, we chose the HepG2 cell model for this pilot study, which is considered useful for investigating the genotoxicity of compounds with unknown or poorly known mechanisms of action [20,27]. 

This research represents the first study that applied such an experimental design with the HepG2 cell model and tanning agents. In the light of real application, such basic in vitro research cannot be directly applied. We are very aware of the limitations and methodological constraints of the model used. We also anticipated potential problems in extrapolating the obtained results to the real exposure scenario and did not intend to assume analogous results in humans. Although results of in vitro studies do not provide definite answers on potential toxic outcomes following the application of some procedure, they could point to the risks associated with exposure to the tested agents which were observed at the lowest, i.e., cellular, level of the biological organisation. Their value is also reflected in the fact that every positive result is basically a warning sign that several other additional tests must be carried out to avoid any risk for the end users of materials processed using potentially harmful procedures. Our experimental design included a 24 h in vitro exposure of HepG2 cells to the leather samples tested. This period was long enough to cause the release of various compounds used for tanning and dyeing from the tested materials into the liquid cell medium RPMI 1640. This was demonstrated by observing visual colour changes of the medium. However, this was not accompanied by pH imbalance, as confirmed by measurements of the pH of the medium after treatment. Unfortunately, because this pilot study did not involve complex chemical analyses, specific detection of chemicals released into the medium or determination of their total amounts, we cannot determine exactly which specific compounds were leached from a particular leather sample. We can only relate the measured genotoxic effects to the known composition of the tanning and dyeing chemicals used in the processing of the leather samples used in this study and make assumptions as to what may have contributed to the results obtained.

To assess the genotoxic effects of chemicals released from leather samples processed by different methods, we incubated HepG2 cells with the leather rectangles cut into equal areas (1 × 1 cm^2^). The effects measured in the exposed cells show that the leaching of potentially genotoxic chemicals from the same area is different; it was highest after vegetable tanning, followed by synthetic tanning and chrome tanning. The fact that vegetable tanning causes high levels of DNA damage in HepG2 cells is disappointing, considering that such leather processing methods are considered more harmless and environmentally friendly. However, from a genotoxicological point of view, the results obtained are not surprising, considering the conditions of cell exposure and the specificity of the model used. Leaching is influenced by the solubility of the chemicals used in the tanning (and dyeing) processes. Since the RPMI 1640 medium used to culture HepG2 cells is a liquid whose properties allow mobilisation of hydrophilic compounds [28], it is possible that many of the components used for vegetable tanning are readily released into the medium after 24 h of incubation due to their chemical properties and also enter the HepG2 cells and cause detectable levels of primary DNA damage. A high level of primary DNA damage measured after 24 h does not only mean a high genotoxic risk. Rather, it is a warning sign indicating that chemicals released from a tested sample have a high potency to produce DNA lesions that can be specifically detected by the alkaline comet assay. However, the genotoxic effects are much more complex and depend not only on the overall extent of primary DNA damage measured, but also on the efficiency of its repair and the persistence of unrepaired lesions. The majority of lesions detected by the alkaline comet assay are single-strand breaks and alkali-labile sites in DNA [24,26,29], which are usually extensively repaired and do not cause significant damage. Other lesions detectable with the comet assay include single-strand breaks associated with incomplete excision repair, double-strand breaks, DNA–DNA and DNA–protein crosslinks [23,24]. Some of the DNA damage detectable by the assay is also due to DNA repair processes causing additional lesions [25].

So, what happened after the HepG2 cells came into contact with vegetable-tanned leather? The increased levels of primary DNA damage measured in these cells were apparently due to complex interactions between highly reactive compounds of natural origin that were efficiently released into the cell medium due to their high solubility. Looking at the composition of the substances used for vegetable tanning, it becomes clear that it is a very complex mixture of potentially reactive chemicals. Despite the widespread assumption that natural compounds generally have better biocompatibility and are less harmful to cells, tissues and organisms, it must be emphasised that many chemicals of natural origin in fact express their dual nature and act as both prooxidants and antioxidants, depending on the concentration [30,31]. As is well known, prooxidant behaviour depends largely on the environment in which these compounds are present. It is also modulated by the presence of O_2_ and some redox-active metal ions such as iron and copper, which can lead to oxidative DNA degradation [31,32]. Therefore, chemicals of natural origin used for vegetable tanning could also be genotoxic, depending on their concentration. Tannins are known to be natural products found in most higher plants. They are usually divided into water-soluble (hydrolysable) polyphenols, such as gallo-tannins and ellagi-tannins, and condensed polyflavonoid tannins, which are rarely hydrolysed [33]. The vegetable tanning process used in the processing of a leather sample examined in the present study used tannins from mimosa, chestnut and quebracho. These are now considered the most important sources of plant tannins for leather production, but also represent industrially available natural sources of polyphenols used for many other purposes [34,35,36]. It has long been known that tannin-related substances induce DNA fragmentation [37]. Therefore, the results of the comet assay performed in our study are highly justified by the existing literature. Tannic acid is one of the most important hydrolysable tannins [33]. As early as the mid-1990s, Bhat and Hadi [38,39] observed that tannic acid causes DNA degradation in the presence of Cu (II) through the formation of reactive oxygen species. The cytotoxic and DNA-damaging effects of this compound were recently confirmed using the same cell model as ours, HepG2 cells. Mhlanga et al. [40] found that tannic acid increased both cell death and DNA fragmentation. These results are consistent with our own observations. Nowadays, synthetic tannins are increasingly used and research in this field is growing. Uddin et al. [41] recently investigated the role of glutaraldehyde in the tanning process compared to conventional chrome tanning. Their results suggest the usefulness of glutaraldehyde, especially in terms of minimising chromium pollution and reducing the generation of toxic waste and its impact on the environment. Our results regarding the genotoxicity of leather processed with synthetic tannins are not very convincing—we observed relatively high DNA damage in HepG2 cells exposed to this tested material. The leather sample obtained with synthetic tanning that we tested in the present experiment was processed with the agents Sellatan P and Sellatan CF based on modified polysulphonic acids and glutaraldehyde. Available literature reports indicate that these compounds can cause primary DNA damage. The main problem with glutaraldehyde is that this compound causes DNA–protein cross-linking [42,43,44]. It is important to note that repair of these lesions leads to DNA excision [42], resulting in additional “repair-related” primary DNA damage that can be detected by the alkaline comet assay. Taken together, previously known mechanisms of DNA damage induced by this specific chemical and the results we obtained in the present study match very well. Here, we must emphasise that the sample of leather processed with the synthetic tanning was used in our research only for the purpose of comparison with other samples. The aim of our research was not to collect evidence in support of this type of leather tanning, especially considering that in the meantime, glutaraldehyde was listed on the Substances of Very High Concern (SVHC) candidate list by the European Chemicals Agency [45]. However, the scientific project of which part is the research described in the present manuscript has been conducted continuously over several years, when the ban of glutaraldehyde was not yet in force. As far as we know, the ban happened in 2021, while our research was conducted mostly prior, or parallel to that ban. This means that at the time when our experiments were conducted, we could not predict that the ban would come into force. It is clear to us that prohibited substances may no longer be used in processing of leather, but for the purpose of experiments it is allowed to compare the effects of prohibited substances with those whose use is permitted. More than 90% of the leather produced worldwide is tanned with chromium [8]. However, doubts remain about the amount of chromium released from leather and its impact on human health and the environment. EU legislation states that “leather products that come into direct, prolonged or repeated contact with the skin shall not be placed on the market if the leather contains chromium (VI) in concentrations equal to or greater than 3 mg/kg” [46]. Thus, if the level of toxic Cr in various goods, garments and footwear is kept very low, the risk to consumers could be acceptable. Our results suggest that chromium tanning leads to slightly less primary DNA damage in HepG2 cells compared to tanning with synthetic and vegetable tanning agents. From a toxicological point of view, this is not surprising, considering that the chromium (III) used for tanning is a less hazardous species than chromium (VI) [47,48]. We assume that under our specific experimental conditions, the total amount of chromium (III) leached from chrome-tanned leather was relatively low compared to the amounts of other reactive substances released from the same surface of the other leather samples. Based on the results obtained, it is possible that the cell exposure conditions did not promote the conversion of the released Cr (III) to the more toxic Cr (VI), which is known to be a strong oxidant in acidic media [49]. However, this does not seem to have been the case, which was also supported by the finding that the pH of the medium did not differ from that of the negative control culture. We also found that dyeing chrome-tanned leather with black dye did not cause additional DNA damage compared to chrome-tanned leather that was not further processed. This result could be related to the negligible leaching of the dye into the culture medium, which was documented by the fact that the colouration of the medium did not change noticeably after 24 h of incubation with the sample of black-dyed leather. As this study was a pilot study with only one experimental scenario, we cannot answer whether longer exposure times could lead to greater leaching of chromium or dye and what effects could be expected under these conditions. This needs to be investigated in future studies with a wider range of exposure times. We should also briefly point out the limitations of the experimental design used. Although the model system used here was suitable for a general assessment of DNA-damaging effects, it unfortunately cannot reproduce real-life exposure well. In fact, no in vitro cell system, not even one using skin cells, can authentically mimic exposure from skin contact with a particular material. This is because the cells are grown in a liquid medium, unlike skin cells, which are organised in a very specific tissue. Leather products are usually in close contact with the surface of human skin, which has specific properties and multiple protective barriers against potentially harmful exposures. Such complex protective barriers are absent in cells maintained in cultures, which are more vulnerable and susceptible to higher levels of DNA damage. In addition, the leaching of potentially harmful substances from the skin surface after direct contact with any material is greatly reduced compared to in vitro conditions because the skin surface is normally relatively dry and the intercellular pores are largely impermeable to chemicals. This is the main difference between the exposure of the cells used in our experimental conditions and the real exposure of skin cells. In addition, the risk of harmful exposure following direct contact with a material containing potentially toxic substances would be greater for hydrophobic chemicals, as the outer stratum corneum of the skin is lipophilic. Therefore, under real conditions, one might expect a slower release of the hydrophilic chemicals than was the case under our exposure conditions. With this experimental model, we also cannot predict the extent to which potentially harmful chemicals will penetrate into the deeper layers of the skin and potentially cause significant toxic effects there.

## 4. Materials and Methods

### 4.1. Leather Samples

The samples of natural (bovine) leather used in this study were processed according to the same procedures before tanning and dyeing. Three samples were semi-processed bovine leathers (labelled CHR-T1, SYN-T and VEG-T) tanned with different tanning agents (chrome, synthetic and vegetable tanning agents), and one black-dyed leather sample (CHR-T2), which was dyed after the tanning process and represented the final product. All tested samples were produced by a Croatian company (PSUNJ factory, Rešetari, Croatia). A detailed description of the different tanning agents used in leather production can be found in Table 1.

### 4.2. Cell Culture

The human hepatoma cell line HepG2 was obtained from the American Type Culture Collection (ATCC) (Manassas, VA, USA). Cells were cultivated in RPMI 1640 medium with antibiotics (penicillin/streptomycin) (Sigma-Aldrich, Steinheim, Germany) and 10% fetal bovine serum (FBS) (Sigma-Aldrich). They were seeded in 25 cm^2^ flasks and maintained in a humidified atmosphere of 5% CO_2_ at 37 °C (Heraeus Hera Cell 240 incubator, Langenselbold, Germany).

### 4.3. Experimental Design

Three independent experiments were performed with HepG2 cell cultures grown for 48 h before the treatments. Samples of leather (cut into rectangles of 1 × 1 cm^2^) were first sterilised with UV light (30 min on each side of the leather sample) and then placed in the flasks containing the cell cultures in fresh complete medium. The cultures were incubated with the tested leather samples for 24 h in a humidified atmosphere of 5% CO_2_ at 37 °C (Heraeus Hera Cell 240 incubator, Langenselbold, Germany). After incubation, the medium containing the leather samples was discarded and the pH was measured. The cells were washed with sterile PBS, trypsin-EDTA was added to each flask, and the flasks were further incubated at 37 °C. After detaching the cells, complete medium was added to inactivate trypsin; the cells were mixed gently with a pipette, transferred to tubes and centrifuged at 800× *g* for 4 min. The supernatant was drained off and small amounts of PBS buffer were added to the precipitate. The resulting cell suspensions were used to prepare agarose microgels.

### 4.4. Alkaline Comet Assay

To assess the extent of primary DNA damage in single cells, we used the alkaline comet assay procedure [50,51,52] with slight modifications [53]. Agarose microgels were prepared on slides pre-coated with 1% normal melting point (NMP) agarose. Duplicate slides were prepared for each experimental point. The first layer of the gel consisted of 0.6% NMP agarose. It was covered by: (1) A mixture of 0.5% low melting point agarose (LMP agarose) and cell samples (V = 15 µL suspension) and (2) the top layer of 0.5% LMP agarose. In our experiments, untreated cells served as negative controls. To prepare positive control slides, we exposed microgels containing untreated cells to 30 µmol/L hydrogen peroxide on ice for 10 min. Hydrogen peroxide was chosen for this purpose because it causes extensive DNA damage that can be detected by the method used. As a rule, the use of a known genotoxic substance as a positive control is recommended in order to obtain a positive reaction with the comet assay. After complete preparation, all slides were further processed in the same way. After polymerisation, the gels were immersed overnight in a lysis buffer (2.5 mol/L NaCl (Kemika, Zagreb, Croatia), 100 mmol/L Na_2_EDTA, 10 mmol/L Tris-HCl, 1% sodium lauroyl sarcosinate, pH 10) containing 1% Triton X-100 and 10% dimethyl sulphoxide (Kemika, Zagreb, Croatia). The next day, slides were washed with distilled water to remove excess salt, immersed in freshly prepared cold denaturing/electrophoresis buffer (300 mmol/L NaOH, 1 mmol/L EDTA, pH > 13) and incubated in the dark at 4 °C for 40 min. The slides were then placed in a horizontal electrophoresis unit (SCIE PLAST, Cambridge, UK) filled with the same buffer. Electrophoresis took 20 min at 0.8 V/cm, 300 mA (power supply Power Pac HC^TM^, BIO-RAD, Hercules, CA, USA) and 4 °C. The slides were then neutralised with three portions of Tris-HCl buffer (0.4 mol/L; pH 7.5). The microgels thus prepared were dehydrated with 70% ethanol for 10 min and with 96% ethanol for 10 min, air dried and stored in tightly closed plastic boxes protected from moisture and light until analysis. After staining with ethidium bromide (20 µg/mL), the microgels were analysed under an epifluorescence microscope (Olympus BX51, Tokyo, Japan) at 200× magnification. Analysis was performed by randomly selecting at least 50 comets per microgel using Comet Assay IV^TM^ software (Instem-Perceptive Instruments Ltd., Suffolk, Halstead, UK). As the experiment was performed in triplicate, a total of 300 individual comet measurements were taken for each experimental point. Two comet descriptors were selected to quantify the extent of DNA damage: %DNA in the tail and tail length (expressed in micrometres). Statistical calculations were performed using Statistica—Data Science Workbench software, version 14 (Licence No. 14.0.0.15; TIBCO Software Inc. 2020; Palo Alto, CA, USA). Basic descriptive statistics were calculated for each experimental point. Then, the Mann–Whitney U test was applied to make comparisons between the treated samples and the controls. Comparisons between the values obtained for the number of comets belonging to a given category with respect to four quartiles were made using Pearson’s χ^2^ test for two-by-two contingency tables [54]. The statistical significance level was set at *p* < 0.05.

## 5. Conclusions

Using the HepG2 cell model and the alkaline comet assay method, we have shown that all tanning methods used in leather processing can have genotoxic effects to some degree. Our experimental design allows a reliable estimation of the genotoxic effects mainly of the hydrophilic compounds present in the leather samples processed with different tanning methods. The results suggest that vegetable and synthetic tanning, perhaps due to their complex composition, cause higher overall primary DNA damage than “normal” chrome tanning. 

Despite all limitations, we believe that the obtained results contribute interesting new information useful for the future safety assessments of the studied tanning agents. These preliminary results may be useful to gain a general insight into the genotoxic potential of the processes used in leather processing and to plan future experiments with more specific cell or tissue models. Future research regarding toxic aspects of vegetable tanning should focus on detailed phytochemical characterisation, explaining of interactions between the components included in the vegetable tanning procedures and determination of the concentration ranges in which their possible harmful effects would be minimised, while maintaining efficiency in leather processing. As the potentially useful cell model, skin cell lines are proposed, of which several commercially available types are available. Last, but not least, besides specific modifications of the comet assay, the use of cytogenetic assays is also advised. Among them, there are two in vitro tests proposed by the OECD: Chromosomal aberration test and micronucleus test. The latter test in its “cytome” version can provide a range of useful information valuable for further risk assessments. 

## Figures and Tables

**Figure 1 molecules-27-07030-f001:**
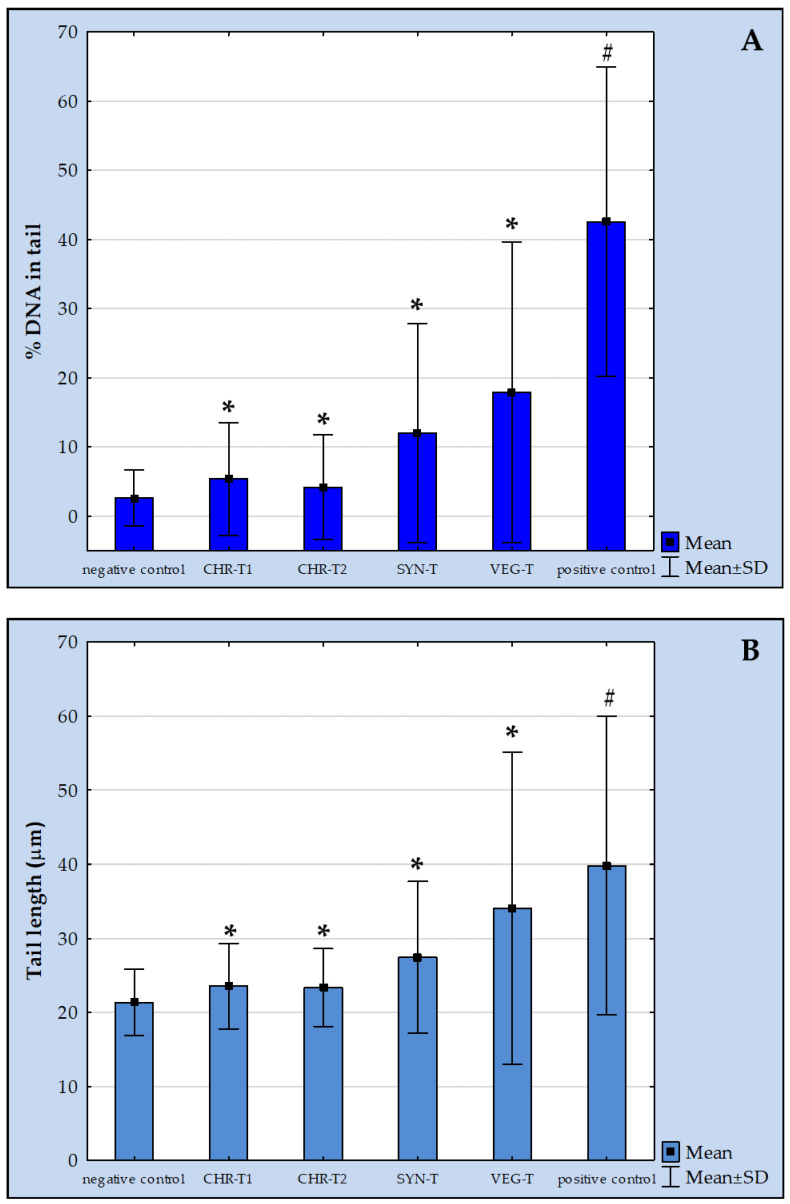
The extent of primary DNA damage in HepG2 cells measured by the alkaline comet assay after 24 h exposure to the leather samples tested and the corresponding negative and positive controls. The main comet descriptors were %DNA in tail (**A**) and tail length (**B**). CHR-T1 and CHR-T2 correspond to the leather sample processed with two chrome tanning procedures; SYN-T corresponds to the leather sample processed with synthetic tanning; VEG-T corresponds to the leather sample processed with vegetable tanning. Results are reported as mean ± standard deviations obtained by the measurements of 300 comets per experimental group. *—significantly increased compared to negative control; #—significantly increased compared to all other experimental groups (*p* < 0.05; Mann–Whitney U test).

**Figure 2 molecules-27-07030-f002:**
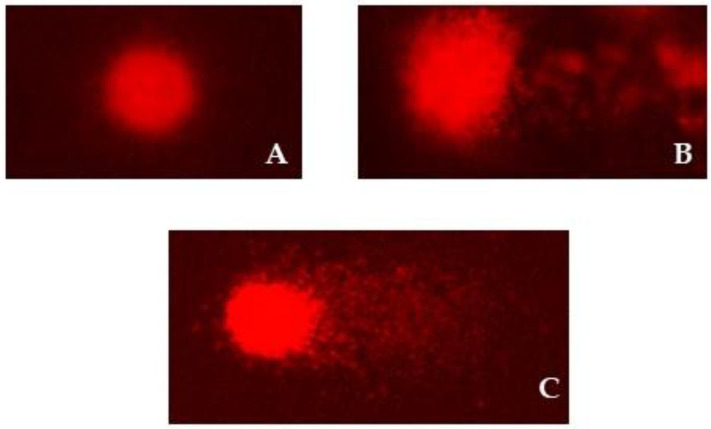
Photomicrographs show nucleoids of HepG2 cells observed in cell cultures after alkaline comet assay procedure: (**A**) Negative control without DNA damage; (**B**) positive control (hydrogen peroxide) with extensive DNA damage; (**C**) damaged DNA after 24 h exposure to leather sample processed with vegetable tanning. The photomicrographs were acquired using Comet Assay IV^TM^ image analysis software (Instem-Perceptive Instruments Ltd., Suffolk, Halstead, UK) at a magnification of 200×.

**Figure 3 molecules-27-07030-f003:**
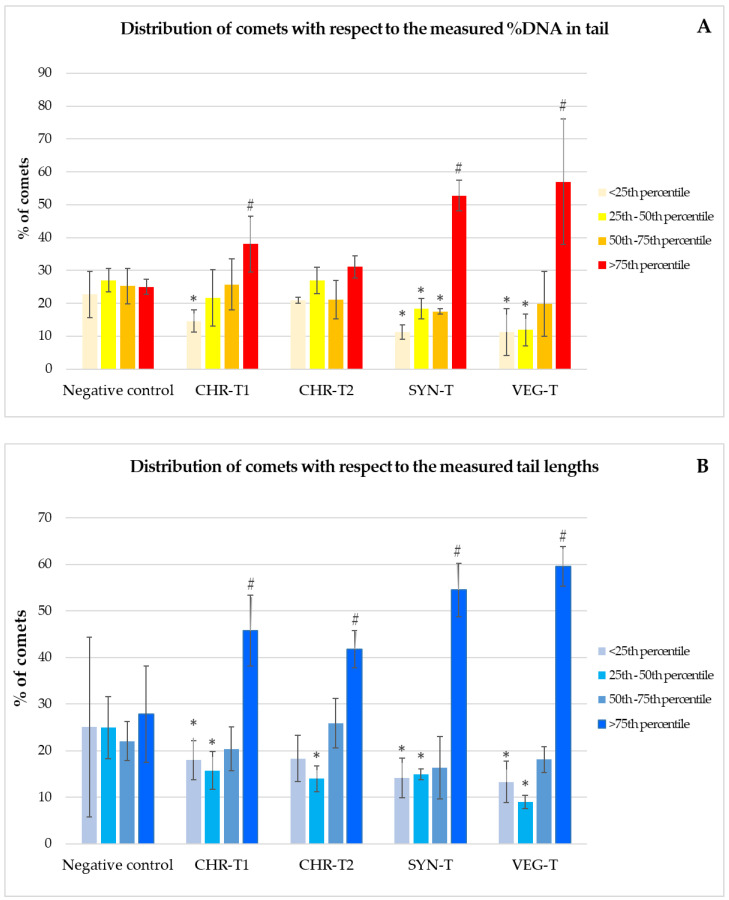
Frequency distribution of DNA damage (**A**,**B**) in HepG2 cells after 24 h exposure to the tested leather samples and the corresponding negative control. CHR-T1 and CHR-T2 correspond to the leather sample processed with two chrome tanning procedures; SYN-T corresponds to the leather sample processed with synthetic tanning; VEG-T corresponds to the leather sample processed with vegetable tanning. Results are reported as mean ± standard deviations of three independent evaluations that each separately included measurement of 100 comets. *—significantly decreased (*p* < 0.05) vs. negative control; #—significantly increased (*p* < 0.05) vs. negative control (Pearson’s χ^2^ test).

**Table 1 molecules-27-07030-t001:** Basic parameters of the technological processes in the tanning of leather with different tanning agents.

Labelling of Leather Samples	CHR-T1	CHR-T2	SYN-T	VEG-T
**Pickling** **process**	1.6–1.8% acid (formic acid, sulphuric acid) 5–7% sodium chloride 40–45% water temp. 19–22 °C pH 3.00	1.6–1.8% acid (formic acid, sulphuric acid) 5–7% sodium chloride 40–45% water temp. 19–22 °C pH 3.00	2.0–2.5% commercial product based on polysulphonic acid, without salts 50% water temp. 20–25 °C	/
**Pre-Tanning process**	/	/	/	7–9% synthetic tanning agent for better tanning process; prevents the reduction of the concentration of the plant extracts) 20% water temp. 20–23 °C pH 5.0–5.5
**Scouring**	/	/	/	water, temp. 30 °C, circular bath
**Tanning process ***	3.2–3.6% basic chromium sulphate (commercially agent 25–27% Cr_2_O_3_; 33^0^Sch) pH 2.4	3.2–3.6% basic chromium sulphate (commercially agent 25–27% Cr_2_O_3_; 330Sch) pH 2.4	1.5–2.0% synthetic tanning agent based on aliphatic polyaldehyde, metal free 2.0–2.5% commercial product based on polysulphonic acid, 0.1–0.2% formic acid (85%), pH 3.5 0.1–0.2% sodium bicarbonate 0.1–0.2% sodium bisulphite, pH 3.8–4.0	4–6% mimosa; 9–11% chestnut; 9–11% quebracho 1.5–2.5% synthetic tanning agent (for softness, suppleness and strength, dyeing) water 20% pH 3.0–3.5
**Basification process**	0.25–0.32% basifying agents (commercial preparations of salt mixtures with low alkaline reactivity)—pH 10–12), water dispersed fungicide with zero volatile organic compounds; 20–25% water temp. 50 °C	0.25–0.32% basifying agents (commercial preparations of salt mixtures with low alkaline reactivity)—pH 10–12), water dispersed fungicide with zero volatile organic compounds; 20–25% water temp. 50 °C	/	/
**Dyeing**	/	Aniline Black dyes	/	/

Note: The quantity of tanning agents and other agents used to process the leather is not constant and is always adjusted depending on the quality of the raw or semi-finished product that is processed into finished leather, the quality of which must meet the buyer’s criteria and is the responsibility of the manufacturer. * Depending on the tanning agent, the technological processes differ in terms of concentration, treatment time, pH, water content and chemical auxiliaries in the tanning bath required to obtain leather with high quality and good properties.

## Data Availability

Not applicable.

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
