# Peer review of "Evaluation of DNA-Damaging Effects Induced by Different Tanning Agents Used in the Processing of Natural Leather—Pilot Study on HepG2 Cell Line"

_molecules, 2022, doi:10.3390/molecules27207030_

Round 1
Reviewer 1 Report
General comment:
This Manuscript analyze the effect of different tanning agents on DNA damage a carried out in leather as media. Extent of damage was assessed by Comet assay. However, there are several points need to be clarified as specific comments as follows:
Specific comments:
· Why n HepG2 cells have been taken and what is the logic.
· What is the amount of different tanning agents used.
· What is Comet assay and how the DNA damage is assessed and how the results are interpreted in the analysis. A brief account is needed with some relevant References.
· What is the effect of pH on the assay and the effect of pH on tanning and their relaton.
· In Fig. 1. It is confusing to see Different colors such as Green, pink etc. It is more useful and understandable to give the exact Tanning agents name may be with some abbreviations.
· Synthetic tanning agents Sellatan P and Sellatan CF as used in the experiments, which contains modified polysulphonic acids and glutaraldehyde as mentioned, whereas, glutaraldehyde is already banned by recent REACH norms. Hence, there is no point in using such Tanning agents.
· In 3. Discussion, which indicates more DNA damage caused by Natural vegetable tanning agents, is higher, but at what pH assay carried out and its effect on Tanning and Tanning agents. DNA is a protein, whereas, Vegetable Tannins are only expected interact with DNA and produce crosslinking.
· What is the procedure adopted for Tanning using different Tanning agents.
· Is there any other Analysis performed apart from Comet assay, such as Molecular weight or SEM / TEM analysis to conform the damage and extent of the damage?
· What is the actual relevance of the present study in the angle of Real application.
· Is there any analysis carried out with only using Different tanning agents and DNA, but without using Leather?
Therefore, Authors should revise the Manuscript as per the Specific comments.
Author Response
Dear Reviewer,
please see the attachment with responses.
Thank you.

Reviewer 2 Report
In the manuscript, the authors performed the detection of DNA damage by Tanning Agents. The manuscript is simple and lacks sufficient experimental data, and the authors are advised to add and resubmit the manuscript. Some issues.
1. it is not enough to demonstrate DNA damage by comet assay alone, the authors need to add other experiments for characterization. Instead of doing only one experiment and performing statistical analysis on it repeatedly.
2. The authors should explain the basis for their choice of the concentration of Tanning Agents used.
3. For the comet experiment, the authors should show all the original images, not just the representative images of Fig2.
4. The authors should explain the basis for the selection of cells.
Author Response

(The authors gave the same response as above.)

Round 2
Reviewer 1 Report
Comments:
· Authors have state, they have used different appropriate amount of different tanning agents used. Ok. But, what exactly the quantity that they have used for Leather, which they have taken for their testing is very important, as this has got possible direct relation with damage effect as per the present study.
· Authors have also stated that the amount of Tanning agent available after Leaching from Leather would affect the damage as per the study. What is the amount of Tanning agents available after Leaching under the given process conditions for causing the damage as studied?
· Is there any standard protocols to assess the damage as Effect of Toxicity or Toxicological studies as per the different agencies such as OSHA, EU, REACH etc.?. I think there should be some Standards available for such Toxicological studies. Please Cite the References and also check the present study fall in line with those standard protocols.
Author Response
Dear Reviewer,
please see the attachment.
With kind regards,
Authors

Reviewer 2 Report
Authors should label all bar graphs with SD values. The paper can be accepted after the authors have made corrections.
Author Response

(The authors gave the same response as above.)
